# Malocclusions in Pediatric Patients with Asthma: A Case–Control Study

**DOI:** 10.3390/healthcare10081374

**Published:** 2022-07-24

**Authors:** Jocelyn Castañeda-Zetina, Martha Gabriela Chuc-Gamboa, Fernando Javier Aguilar-Pérez, Alicia Leonor Pinzón-Te, Iván Daniel Zúñiga-Herrera, Vicente Esparza-Villalpando

**Affiliations:** 1Faculty of Dentistry, Autonomous University of Yucatán, Calle 61-A No. 492-A, Mérida 97000, Mexico; jocy_castan_94@hotmail.com (J.C.-Z.); martha.chuc@correo.uady.mx (M.G.C.-G.); alicia.pinzon@correo.uady.mx (A.L.P.-T.); ivan.zuniga@correo.uady.mx (I.D.Z.-H.); 2Faculty of Stomatology, Autonomous University of San Luis Potosí, Av. Dr. Manuel Nava No. 2, Zona Universitaria, San Luis Potosí 78290, Mexico; vicente.med@outlook.com

**Keywords:** asthma, malocclusions, pediatric patients, oral habits, tongue habits

## Abstract

Asthma is a public health problem that has been widely described, but little has been reported about its effects on dental occlusions. The aim of this study was to compare the alterations of normal occlusions in asthmatic children and those without the disease. The study included 186 patients between 5 and 12 years old, divided into two groups. The first group included patients with a previous diagnosis of asthma given by a specialist, which was confirmed by using the International Study of Asthma and Allergies in Childhood questionnaire. The second group included patients without the disease. All patients underwent a clinical examination to determine the presence of occlusion alterations in the sagittal, transverse, and vertical planes. Subsequently, chi-squared tests were performed to compare the variables between the groups. A significant association was found between asthma and the variables studied here: alterations in the sagittal plane (chi^2^ = 7.839, *p* = 0.005), alterations in the vertical plane (chi^2^ = 13.563, *p* < 0.001), alterations in the transverse plane (Fisher’s F *p* < 0.001), and oral habits (chi^2^ = 55.811, *p* < 0.001). The results suggest that asthmatic patients are more likely to develop malocclusions, especially anterior open bite and posterior crossbite. These conditions are typically related to mouth breathing, which is common in asthmatic patients.

## 1. Introduction

Different diseases or conditions related to the respiratory system are common and have negative impacts on quality of life and life expectancy. In such cases, one of the basic vital needs of all human beings is compromised—breathing. Breathing enables the movement of oxygen from the outside environment to the cells within tissues, which is necessary for cell function, the release of waste products, and preventing cell death [1,2].

Asthma is a disease that affects individuals of all ages all over the world, and the World Health Organization (WHO) has established it as a public health problem [3]. This disease is characterized by inflammation of the airways that makes breathing difficult by obstructing the passage of air. Consequently, the body adapts to this insufficiency, which could affect normal craniofacial development [4]. Its etiology has not been established, since there are several factors involved that favor its development. The most common symptoms are wheezing, coughing, and dyspnea. It can completely or partially revert in a spontaneous way or with the use of medications [5].

The WHO estimates that more than 300 million people around the world currently suffer from asthma, and at least 250,000 deaths are attributed to it each year. In pediatrics, asthma is considered the most frequent chronic respiratory disease, and although the prevalence varies for each country, it is estimated that the range in children and adolescents is approximately 2 to 25%. The International Study on Asthma and Allergies in Childhood (ISAAC) is considered one of the most important studies worldwide and the main source of information on the prevalence of asthma. The study found that the prevalence of this disease in children in Mexico ranges between 5% and 14%, with an average of 8% and higher prevalence in cities near the Gulf of Mexico. In Yucatán, particularly in the city of Mérida, epidemiological studies have reported a prevalence of 12% in children and an incidence of 7.75 ± 0.15 per 1000 population [6,7,8,9].

Regarding the oral health of asthmatic children, previous studies have demonstrated a strong association between dental caries and asthma. Asthmatic children presented higher values in the DMF index, worse gingival health, disorders of salivary flow, saliva composition, and decreased saliva pH [10,11,12,13]. There is evidence that the oral microbiota of children with asthma is colonized by opportunistic bacteria with high pathogenic capacity, such as *Streptococcus, Neisseria, Veillonella, Prevotella, Haemophilus, Kingella, and Porphyromonas*, which are possibly involved in the pathogenesis of both caries and asthma [10]. In addition, there are changes in the bucco-nasal-pharyngeal microbiota caused by mouth breathing, which stimulate the presence of certain pathogens, accompanied by alterations in the salivary environment, which could play an important role in developing periodontal affection [12,14]. Mouth breathing is due to abnormal craniofacial development and can cause serious systemic problems in adulthood. It is considered a risk factor for the development of atopic dermatitis and may be a risk factor for tonsillitis (tonsillar hypertrophy) and class II dental malocclusions [15]. The early detection of pathognomonic signs of mouth breathing makes it possible to treat children with an early multidisciplinary approach, allowing them adequate physical and psychological development [16].

The American Academy of Pediatric Dentistry indicates that the risk for oral health conditions may be higher among children with chronic conditions (such as asthma) who take medication. Many studies over the years have described oral manifestations of pediatric asthmatic patients, such as caries, dental erosion, malocclusions, gingival disease, xerostomia, and candidiasis, among others [4,17]. The physiology of asthmatic patients’ needs to adapt to compensate for their respiratory disability. Some of these adaptations involve posture changes at the cervical level, which have been reported to affect the normal growth pattern and development of the craniofacial complex [17]. A high frequency of oral breathing has also been reported in asthmatic patients, which produces changes in the maxillary bones, oral muscles, facial muscles, and the dento-alveolar complex in general. These conditions usually develop into dental malocclusions [18,19]. The aim of the present study was to compare the alterations of normal occlusions in pediatric asthmatic patients to those without the disease.

## 2. Materials and Methods

This observational, comparative, cross-sectional study used a case–control design. The study was reviewed and approved by the research committee of the Faculty of Dentistry of the institution responsible for the project with registration number SISTPROY FODO2018-0003. It was carried out under the ethical principles of the Declaration of Helsinki, and the personal data of all participants were kept strictly confidential.

The population consisted of patients between 5 and 12 years old who attended a public institution that provides dental care in Southeastern Mexico from September 2018 to February 2019. The sample selection was non-random based on convenience. In order to obtain reliable results, patients were excluded if they had any other respiratory or systemic disease other than asthma, had a developmental syndrome or disorder, craniofacial malformations, a history of dental trauma, or had used dental orthopedic appliances.

To carry out the data collection, pertinent questions were first asked to allow for the identification of the established inclusion criteria. Furthermore, since the participants were minors, consent for their participation was obtained from their parents or guardians, who signed a corresponding document, and then the minors agreed to participate. Asthmatic patients were considered as those whose guardian stated that they had received a previous diagnosis from a specialist doctor. Additionally, the asthma diagnosis was confirmed using a questionnaire that was proposed and validated by the ISAAC [20].

Each individual underwent an intraoral clinical examination following pertinent protection measures dictated by Official Mexican Standard NOM-013-SSA2-2016. The instrument for the analysis of occlusions used for data collection was completed with the help of a tongue depressor and a millimeter ruler. This instrument was designed using the points of interest for the investigation and was supervised and approved by an orthodontic specialist and a pediatric dentist. The analysis of the occlusions included an assessment of the molar relationship or terminal plane as appropriate, as well as the presence of alterations of normal occlusions, such as anterior crossbite, anterior open bite, posterior crossbite, and oral habits.

For the analysis, alterations in normal occlusions were grouped into the following categories: (A) Alterations in the sagittal plane were defined by the presence of anterior crossbite, class II and III molar relationships (in patients with erupted and occluded first permanent molars), or an exaggerated mesial or distal step in terminal plane (in patients with incomplete eruption of the first permanent molars). The class I molar relationship and the flush and mesial step terminal planes were considered normal occlusions, so they were not included as occlusion alterations. (B) Alterations of the transverse plane were defined by the presence of bilateral or unilateral posterior crossbite. (C) Alterations of the vertical plane were defined by the presence of anterior open bite. (D) Oral habits included finger sucking, atypical swallowing, lip sucking, and onychophagia.

The information for each patient was kept confidential for later analyses, and no minor could be identified. The sample was divided into two groups, patients with a diagnosis of asthma and patients without a diagnosis of asthma, and the ratio of the numbers of individuals was 1:2. The data were analyzed with descriptive statistics. Chi-squared statistical tests were used to analyze the asthma variable with respect to the variables of anterior open bite, sagittal plane malocclusions, and oral habits. Fisher’s F test was used to analyze the variable of posterior crossbite with respect to asthma.

## 3. Results

The sample consisted of 186 patients between 5 and 12 years of age with and without asthma at a 1:2 ratio. The mean age of the sample was 7.6 years with a standard deviation of ±1.9 years, and the median and mode were 7 years. Sex data are presented in Figure 1.

The frequency rates of occlusion alterations in the studied groups are presented in Table 1 and the results of the statistical analyses are presented in Table 2. In the sagittal plane, asthmatic patients presented the same frequency rates of molar relationship class I (44.2%, *n* = 19) and class II (44.2%, *n* = 19) and a lower frequency for class III (11.6%, *n* = 5). In the case of healthy patients, the most frequent moral relationship was class I (59.8%, *n* = 58), followed by class II (33.0%, *n* = 32), while the least frequent was class III (7.2%, *n* = 7). Regarding the transverse plane, only 9 of the total of 186 patients presented posterior crossbite, and all corresponded to patients with a diagnosis of asthma. Thus, the frequency of this malocclusion in pediatric patients with asthma was 14.5%, and it did not occur in healthy patients. Fisher’s F distribution was used to establish a relationship between asthma and malocclusions in the transverse plane, and the results showed that the association between these variables was significant (*p* = 0.00003).

In the vertical plane, 21.0% (*n* = 13) of the 62 patients in the sample with a diagnosis of asthma presented an anterior open bite, while only 4.0% (*n* = 5) of the 124 healthy patients had this malocclusion. In the whole sample, 47 patients with asthma and 24 healthy patients presented at least one oral habit, meaning that the frequencies of oral habits were 75.8% and 19.4%, respectively (Figure 2).

## 4. Discussion

The results of this study showed that a significant association was found between asthma and alterations in the sagittal plane (chi^2^ = 7.839, *p* = 0.005), alterations in the vertical plane (chi^2^ = 13.563, *p* < 0.001), alterations in the transverse plane with value (Fisher’s F *p* < 0.001), and oral habits (chi^2^ = 55.811, *p* < 0.001).

In terms of the molar relationships, the results of this research agree with those reported by other authors. For example, García et al. clinically analyzed patients with respiratory disease and reported that the class I molar relationship (61%) was the most common, followed by class II (31%) and class III (8%) [21]. On the other hand, Silva et al. studied a population made up of children with a mouth-breathing habit and found that the most frequent molar relationship was class II (60%), followed by class I (36.9%) and class III (3.1%) [22], differing from the present study. These differences may be due to the characteristics of the individuals in each region.

Regarding the transverse and vertical planes, in 2012 Kumar and Nandlal studied a sample composed of 44 healthy patients and 44 patients with asthma between 6 and 12 years old. They found high prevalence rates of anterior open bite and posterior crossbite. The frequency rates for the group of children aged 6–8 years were 46% for anterior open bite and 20% for posterior crossbite. For the age range of 10–12 years old, similar values of 50% and 30% were obtained, respectively. Patients who used inhaled medications for asthma control had a higher frequency of malocclusions [23]. In our study, no comparisons were made between patients who used inhaled medications and those who did not.

In 2017, Ramos et al. reported on a sample made up of pediatric patients with asthma, of which 64.3% presented an anterior open bite [4]. Like Aguilar in 2011, they found a statistically significant relationship between oral breathing and anterior open bite (*p* < 0.0001) [24]. The results in the present investigation coincide with publications presented by multiple authors when establishing an association between asthma and the presence of anterior open bite [4,24]. In our sample, the group of asthmatic patients presented a significant frequency of anterior open bite, which was the most common malocclusion in this group.

Regarding the relationship between asthma and the presence of mouth breathing habits, the results of our study indicated that 46.7% of patients with asthma presented this habit, which coincides with the result reported by Lopes dos Santos et al. in 2012. Using a chi-squared statistical test, they found an association between asthma and mouth breathing (*p* < 0.001), with 66.3% of children with asthma suffering from this harmful habit [25]. Other authors such as Stensson et al. [26] support Lopes dos Santos. When comparing a group of healthy children with one of children with asthma, they established that asthmatics presented the habit of oral breathing twice as much as the control group [25,26]. Additionally, in a systematic review and meta-analysis by Araujo et al., 12,147 subjects were examined, including 2083 children. They reported a significant association between oral breathing and asthma in children (OR 2.46, 95% CI 1.78–3.39) [18]. The mentioned studies support the significant association between asthma and the habit of oral breathing found in this research.

The results found in this study coincide with what has been previously mentioned about progressive craniofacial growth. During childhood, the growth and development process is still active. For the desired and ideal size, shape, and position of the structures involved to be achieved, there should be no interferences that alter this process. One of these interferences could be the obstruction of the airways, since nasal breathing is an important and fundamental part of the correct development of the face and has a particular impact on the oral cavity, with which it is closely related. Nasal breathing is not only important for this aspect, but it is essential for life to obtain oxygen. Therefore, when there is an anatomical or physiological situation that prevents this process, as occurs in asthma, this function is modified, which has important repercussions on skeletal and dental growth [4,27]. For example, it is reported that asthma can affect the pH of the enamel matrix because of the respiratory acidosis and abnormal oxygen levels associated with asthma, potentially developing dental enamel defects [28].

The scientific evidence supports that there is an association between oral breathing and respiratory diseases such as asthma. When there is difficulty in breathing, oral respiration arises not only as a habit, but as an adaptive function. Well-established skeletal and dental characteristics have been reported for such patients with oral respiration [4].

One of the adaptations found in oral breathers is a postural change of the head in order to facilitate the entry of air by placing it in a forward position with a slight backward tilt. These postural modifications produce changes in the mandibular and lingual position and an imbalance in the perioral muscles. At the mandibular level, there is a posterior rotation of the mandible that favors the appearance of skeletal or dental angle class II [29,30].

Furthermore, an oral breather must keep the lips slightly parted with a lower position of the tongue to favor the passage of air into the airway. Thus, constant pressure and air resistance are exerted on the hard palate, creating a negative pressure that stimulates the growth of the lower third of the face in the vertical direction. Without the stimulation of the tongue, this stops the transverse growth in the ogival and maxillary palates, which can lead to posterior crossbites, crowding, and ectopic eruptions, e.g., buccally displaced canines are more likely to be found in patients with a smaller palatal surface area and volume [31]. On the other hand, a low and anteriorized position of the tongue favors atypical swallowing. Thus, an anterior open bite results from constant pressure being exerted against the palatal surfaces of the upper incisors, which generates buccal inclination [4,29,30].

Additionally, the presence of habits associated with asthma (e.g., pacifier use, thumb sucking, nail biting, mouth breathing, among others) can lead to an altered pattern of muscle activity, increasing the risk of disorders in swallowing, phonation, and breathing and changes in dental arches and occlusions, meaning early detection and intervention to prevent or eliminate these habits in asthmatics are crucial, since they have a direct impact on the quality of life of children with this condition [32].

These findings can contribute to improving the quality of life of children with asthma by carrying out preventive and interceptive measures that will favor correct craniofacial growth and development, leading the dental–skeletal complex in harmony to avoid the development of malocclusions, especially those that have been associated anterior and posterior crossbites and their complications.

## 5. Conclusions

The results of the present study suggest that pediatric patients with asthma are related with oral habits, especially mouth breathing. Combined with the characteristics of the disease, mouth breathing acts as a predisposing factor to the development of some alterations of normal occlusions, particularly posterior crossbite and anterior open bite.

Chronic diseases such as asthma require multidisciplinary and interdisciplinary assessments in order to establish a specific approach for these children. It is suggested that asthmatic patients be monitored during their growth and development to prevent and promptly treat possible alterations in occlusions.

## Figures and Tables

**Figure 1 healthcare-10-01374-f001:**
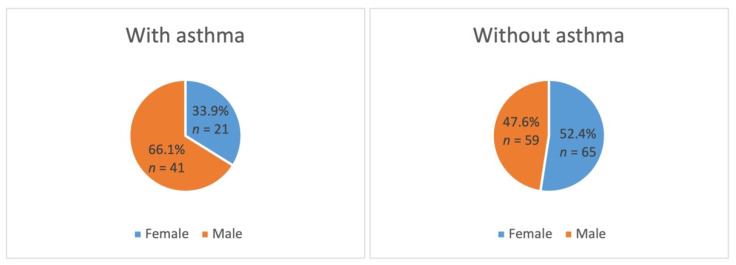
Distribution of the sample by sex.

**Figure 2 healthcare-10-01374-f002:**
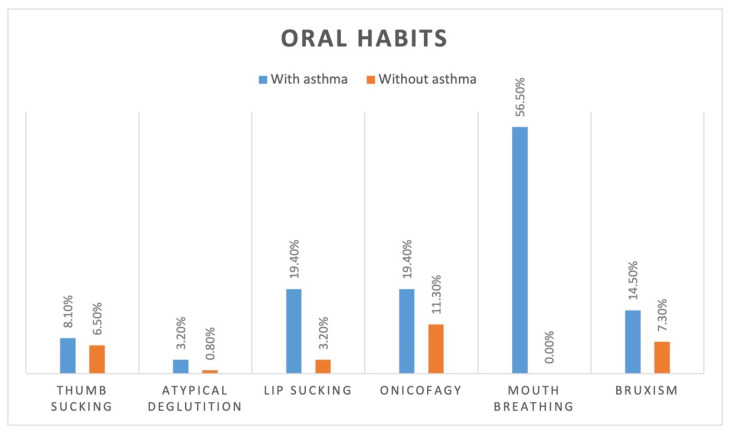
Frequency of oral habits by group.

**Table 1 healthcare-10-01374-t001:** Frequency of occlusion alterations in the studied groups.

Group	Frequency *n* (%)			
	Sagittal Plane	Transverse Plane	Vertical Plane	Oral Habits
**WITH ASTHMA**	39 (90.7)	9 (14.5)	13 (21.0)	47 (75.8)
**WITHOUT ASTHMA**	86 (71.7)	0 (0)	5 (4.0)	24 (19.4)

**Table 2 healthcare-10-01374-t002:** Results of the statistical analyses for asthma and each variable studied.

Variable	X^2^	*p*	OR	95% CI
Sagittal plane	7.839	0.005 *	2.424	1.241–4.826
Transverse plane	-	<0.001 *^,^^†^	-	-
Vertical plane	13.563	<0.001 *	6.244	1.959–23.605
Oral habits	55.811	<0.001 *	12.817	5.936–29.164

* Statistical significance. ^†^ F-Fisher *p*-value. OR = odds ratio; CI = confidence interval.

## Data Availability

The data presented in this study are available on request from the corresponding author.

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
