# Peer review of "Malocclusions in Pediatric Patients with Asthma: A Case–Control Study"

_healthcare, 2022, doi:10.3390/healthcare10081374_

Round 1

Reviewer 1 Report

Dear Authors,

The aim of the present study was to compare the alterations of normal occlusion in pediatric asthmatic patients to those without the disease. The results of the present study suggest that pediatric patients with asthma are at higher risk of presenting oral habits, especially mouth breathing.

The study is of scientific interest and in line with the aims of the journal, but the Discussion section should be improved.

Materials and methods

-       When patients were recruited?

-       Lines 76-77. It was not reported if patients with craniofacial malformations (including cleft lip or palate), history of dental trauma, oral neoformations and other oral cavity pathologies were excluded from the study. Please clarify the inclusion and exclusion criteria. 

-       Lines 84-85. Please add reference.

Discussion

-       Please start the discussion by briefly reporting the salient points of the results.

-       In my opinion, it would be useful to discuss briefly why it could be useful to treat: a) molar relationship (for example the greater the OVJ the greater the incisal trauma); b) malocclusion on the transversal plane (expansion could be beneficial in improving respiratory symptoms - Yoon et al. doi: 10.1016/j.sleep.2022.02.011; reducing the risk of impacted canine – please cite Bizzarro et al. doi: 10.1093/ejo/cjy023; improving the functionality of the masticatory system, excetera..); c) the beneficial of open bite and d) oral habits resolution.

-       Lines 164-166. Please add references.

-       Lines 182-189. Please add references.

Reviewer 2 Report

The manuscript presents original concepts that lead to preliminary evaluations in the event of a pediatric patient suffering from asthma present on the first visit.

before proceeding with the evaluation for publication, the need for a major revision:

Well described abstract

Keywords: Few, others need to be added

Introduction: how does the oral microbiota change in asthma patients? How does the buffering effect of saliva change by having oral breathing? What is the indication of caries

Materials and methods: it should be added which instruments they used at home for oral hygiene and which ones were recommended.

Results well described

Discussions: to add as future objectives, to maintain a state of homeostasis of the oral cavity, through the use of probiotic and post biotic paraprobiotics for periodontal tissues, and the use of biomemental hydroxyapatite for the remineralization of hard tissues. Both of these categories were studied by Prof. Scribante's research group.

Conclusions; to add proactive action

Bibliography: add references from the introduction and discussions.

Round 2

Reviewer 1 Report

Dear Authors,

In my opinion the manuscript is suitable for publication.

Reviewer 2 Report

The manuscript was correctly revised according to the suggestions made in the first revision